# Severe and mild-moderate SARS-CoV-2 vaccinated patients show different frequencies of IFNγ-releasing cells: An exploratory study

Eugenio Garofalo[1☯], Flavia Biamonte[2,3☯], Camillo Palmieri[2], Anna Martina Battaglia[2], Alessandro Sacco[2], Eugenio Biamonte[1], Giuseppe Neri[1], Giulio Cesare Antico[4], Serafina Mancuso[4], Giuseppe Foti[5], Carlo Torti[6], Francesco Saverio Costanzo[2,3], Federico Longhini[1]*, Andrea Bruni[1]

1 Department of Medical and Surgical Sciences, Anesthesia and Intensive Care Unit, University Hospital Mater Domini, Magna Graecia University, Catanzaro, Italy, 2 Department of Experimental and Clinical Medicine, "Magna Graecia" University of Catanzaro, Catanzaro, Italy, 3 Interdepartmental Center of Services (CIS), Molecular Genomics and Pathology, "Magna Græcia" University of Catanzaro, Catanzaro, Italy, 4 Unit of Biochimica Clinica, University Hospital Mater Domini, Catanzaro, Italy, 5 Unit of Infectious Disease, Grand Metropolitan Hospital, Reggio Calabria, Italy, 6 Department of Medical and Surgical Sciences, Unit of Infectious and Tropical Diseases, "Magna Graecia" University, Catanzaro, Italy

☯ These authors contributed equally to this work.
* longhini.federico@gmail.com

**Data Availability Statement:** All relevant data are within the paper.

## Abstract

### Background

Despite an apparent effective vaccination, some patients are admitted to the hospital after SARS-CoV-2 infection. The role of adaptive immunity in COVID-19 is growing; nonetheless, differences in the spike-specific immune responses between patients requiring or not hospitalization for SARS-CoV-2 infection remains to be evaluated. In this study, we aim to evaluate the spike-specific immune response in patients with mild-moderate or severeSARS-CoV-2 infection, after breakthrough infection following two doses of BNT162b2 mRNA vaccine.

### Methods

We included three cohorts of 15 cases which received the two BNT162b2 vaccine doses in previous 4 to 7 months: 1) patients with severe COVID-19; 2) patients with mild-moderate COVID-19 and 3) vaccinated individuals with a negative SARS-CoV-2 molecular pharyngeal swab (healthy subjects). Anti-S1 and anti-S2 specific SARS-CoV-2 IgM and IgG titers were measured through a chemiluminescence immunoassay technology. In addition, the frequencies of IFNγ-releasing cells were measured by ELISpot.

### Results

The spike-specific IFNγ-releasing cells were significantly lower in severe patients (8 [0; 26] s.f.c.×$10^6$), as compared to mild-moderate patients (135 [64; 159] s.f.c.×$10^6$; p<0.001) and

**Funding:** The authors received no specific funding for this work.

**Competing interests:** There are no conflicts of interest related to the present work. Dr. Longhini contributed to the development of a new device not discussed in the present study (European Patent number 3320941 released on 5th August 2020) and he is designed as inventor. He also received speaking fees from Intersurgical, Draeger and Fisher & Paykel. The remaining authors have no conflict of interest to disclose.

healthy subjects (103 [50; 188] s.f.c.$\times 10^6$; p<0.001). The anti-Spike protein IgG levels were similar among the three cohorts of cases (p = 0.098). All cases had an IgM titer below the analytic sensitivity of the test. The Receiver Operating Curve analysis indicated the rate of spike-specific IFNγ-releasing cells can discriminate correctly severe COVID-19 and mild-moderate patients (AUC: 0.9289; 95%CI: 0.8376–1.000; p< 0.0001), with a diagnostic specificity of 100% for s.f.c. > 81.2 x $10^6$.

## Conclusions

2-doses vaccinated patients requiring hospitalization for severe COVID-19 show a cellular-mediated immune response lower than mild-moderate or healthy subjects, despite similar antibody titers.

## Introduction

Infection by Severe Acute Respiratory Syndrome CoronaVirus 2 (SARS-CoV-2) is characterized by the development of a complex disease (COVID-19) with a wide range of respiratory [1] and non-respiratory [2–4] symptoms, which may lead to critical illness and death of patients [5]. The immune system reacts to the virus through the innate and adaptive responses. The innate system reacts to SARS-CoV-2 by recruiting specialized immune cells, such as plasmacytoid dendritic cells and macrophages, whereas the adaptive immune system mainly comprises antibody-producing B cells, and CD4$^+$ and CD8$^+$ T cell endowed with helper and effector functionalities (CD4$^+$) [6, 7]. Both humoral and cellular components of adaptive immunity play distinct and complementary roles in disease resolution and protection from infection or re-infection. In an elegant study by Sekine et al. [8], SARS-CoV-2-Specific T-Cells have been characterized in acute and convalescent unvaccinated patients with asymptomatic to mild disease. The authors reported that, while in early acute phase of SARS-CoV-2 infection CD8+ T cell populations mainly expressed immune activation and cytotoxic molecules together with inhibitory receptors, in the convalescent phase SARS-CoV-2 specific T-cells were skewed toward an early differentiated memory phenotype [8]. Therefore, the time from virus exposure determined the emergence of specific memory cells against SARS-CoV-2 [8]. Another study conducted in unvaccinated patients further confirmed that SARS-CoV-2 specific T-cells were present also during asymptomatic SARS-CoV-2 infections, with a similar initial Interferon-Gamma (IFNγ) secreting T-cell count to severe COVID-19 patients [9].

Anti-SARS-CoV-2 vaccines are the most important preventive strategy against critical forms of COVID-19 [10]. In the beginning, the vaccination with theBNT162b2 vaccine consisted of two consecutive mRNA doses administered 21 days apart [11]. The first cycle of vaccination induces a spike-specific humoral and cellular immune response that was shown to be efficient in 95% of naïve individuals [12]. However, the immune response induced by the dual doses of BNT162b2 vaccine wanes over a period of months, deeming necessary a "booster" dose [10, 13, 14], particularly in immunosuppressed patients [15].

In patients with advanced age or comorbidities, hospitalization and critical COVID-19 have been reported even if recently vaccinated [16] and the efficacy of vaccination wanes over time [14]. These patients presented a low whole blood IFNγ release, despite high anti-Spike IgG titers [16]. Another study suggests that the BNT162b2 mRNA vaccine provides a different (i.e., poorer) immune response (including IFNγ-secreting T-cell counts) in older COVID-naïve adults, as compared to younger cases [17].

Despite the growing knowledge on the role of adaptive immunity in COVID-19, from a clinical point of view there remains the urgency to define correlates of protection or risk of severe disease in vaccinated patients. In this study, we aim to evaluate the spike-specific immune responses in patients with severe or mild-moderate SARS-CoV-2 infection, after breakthrough infection following two doses of BNT162b2 mRNA vaccine.

## Materials and methods

After obtaining local Ethical Approval (approval number 54/2022, 17[th] February 2022) and in accordance with the Declaration of Helsinki and the principles of the Good Clinical Practice guidelines, we conducted this prospective cohort study from April to May 2022, including adults which received the two BNT162b2 vaccine doses in previous 4 to 7 months. The trial was prospectively registered on clinicaltrials.gov on 19[th] April 2022 (registration number: NCT05338736). Written informed consent was obtained from all participants. All individual, de-identified datasets generated during and/or analyzed during the study are available from the corresponding author on reasonable request.

### Population

All consecutive adult (≥18 years/old) conscious patients were screened at the emergency department and/or hospital admission if referred to our center from other hospitals without COVID-19 wards.

We enrolled 3 cohorts of 15 consecutive cases: 1) vaccinated patients with a positive SARS-CoV-2 molecular pharyngeal swab with severe to critical COVID-19, as defined by a peripheral oxygen saturation <94% in room air and therefore requiring hospitalization (severe patients); 2) vaccinated patients with a positive SARS-CoV-2 molecular pharyngeal swab with pauci- or asymptomatic COVID-19 not requiring hospitalization (mild-moderate patients) and 3) vaccinated individuals with a negative SARS-CoV-2 molecular pharyngeal swab (healthy subjects). Patients were classified in the 3 cohorts at the time of swab result. All SARS-CoV-2 infected patients must test positive for their first time to be included, whereas healthy subjects should never test positive before. Furthermore, all 45 subjects must have received the two BNT162b2 vaccine doses in the previous 4 to 7 months.

We excluded all cases with one or more of the following criteria: 1) active malignancy; 2) immunosuppressive or immunomodulant therapies; 3) organ transplantation; 4) steroid therapy since more than 10 days; 5) pregnancy; and 6) refusal to participate.

### Data collection and analysis

After inclusion, demographic and clinical features were recorded. Furthermore, blood samples were collected in BD Vacutainer plasma tubes containing lithium heparin tubes (Becton Dickinson; Plymouth, UK).

Chemiluminescence immunoassay technology was used for the quantitative determination of anti-S1 and anti-S2 specific trimeric IgG antibodies to SARS-CoV-2 (LIAISON SARS-CoV-2 TrimericS IgG Assay, DiaSorin, Saluggia, Italy) and for SARS-CoV-2-specific IgM (LIAISON SARS-CoV-2 IgM Assay, DiaSorin, Saluggia, Italy), according to the manufacturer's instructions [18]. IgG levels ≥ 13.0 AU/mL and IgM levels ≥ 1.1 INDEX were considered positive.

The frequencies of IFNγ-releasing cells were measured by Enzyme-Linked immunoSPOT (ELISpot) Plus Human IFNγkit (Mabtech AB, Stockholm, Sweden) following the manufacturer's instructions [19]. Briefly, freshly isolated Peripheral Blood Mononuclear Cells (PBMCs) from whole blood samples were rested overnight before assay. The wells of a microplate precoated with the anti-IFN-γ monoclonal antibody mAb1-D1K were washed with Phosphate

Buffered Saline 1X (Sigma Aldrich, St. Louis, MO, USA) and blocked with culture medium containing 10% batch tested Fetal Bovine Serum (Sigma Aldrich, St. Louis, MO, USA). As standard, $4 \times 10^5$ PBMCs were seeded per well and stimulated with SARS-CoV-2 Spike peptide pool (Mabtech AB, Stockholm, Sweden) at a concentration of 2 μg/ml for 14 h. Negative (i.e., PBMCs treated with peptide vehicle DMSO) and positive (PBMCs stimulated with monoclonal anti-CD3-2) controls were also included. After washing, the wells were developed with human biotinylated IFNγ detection antibody (1:2,000, clone 7-B6-1), followed by incubation with streptavidin-Alkaline Phosphatase and 5-bromo-4-chloro-3'-indolyphosphate p-toluidine salt/nitro-blue tetrazolium chloride-plus substrate. Spots were counted using an automated spot analyzer (BIOREADER-3000, Bio-Sys GmbH, Germany). Mean spot counts for negative control wells were subtracted from the mean of test wells to generate normalized readings, presented as spot forming cells (s.f.c.) $\times 10^6$. All laboratory tests were performed in accredited laboratories of our University Hospital.

## Statistical methods

Given the exploratory design of this study, we arbitrarily enrolled 15 cases per cohort. We assumed all our data as non-parametric. Continuous variables were compared with the Mann-Whitney tests for analysis of variance by ranks. Post-hoc Dunn's test was applied for pairwise multiple comparisons when indicated. The Receiver Operating Curve (ROC) for the rate of spike-specific IFNγ-releasing was designed and the Area Under the Curve (AUC) was computed to assess the ability to discriminate severe from mild-moderate patients. We considered significant two-sided p values <0.05. Statistical analysis was performed using the Sigmaplot v. 12.0 (Systat Software Inc., San Jose, California).

## Results

Demographic, anthropometric, and clinical features of the cases are reported in Table 1.

No patients received steroids before study inclusion and none of the mild-moderate patients deteriorated to severe. All patients had their first positive SARS-CoV-2 molecular pharyngeal swab and symptoms within 36 hours before inclusion.

Fig 1 depicts the frequencies of spike-specific IFN-γ releasing cells (on the left) and the titers of IgG (on the right) measured in our cohorts of cases. In addition, a representative IFN-γ ELISPOT from one case per cohort is shown in Fig 2. The spike-specific IFNγ-releasing cells were

**Table 1. Population characteristics.**

| | Severe patients (n = 15) | Mild-moderate patients (n = 15) | Healthy subjects (n = 15) | p value |
|---|---|---|---|---|
| *Age (years)* | 62 [52; 73] | 53 [51; 60] | 51 [46; 59] | 0.080 |
| *Female sex—n (%)* | 8 (53%) | 9 (60%) | 5 (33%) | 0.315 |
| *Vaccine last dose (days)* | 154 [150; 181] | 154 [151; 177] | 156 [152; 171] | 0.886 |
| *Comorbidities–n (%)* | | | | |
| *Arterial hypertension* | 12 (80%) | 8 (53%) | 7 (47%) | 0.143 |
| *Diabetes* | 4 (27%) | 2 (13%) | 3 (20%) | 0.660 |
| *Obesity* | 4 (27%) | 2 (13%) | 2 (13%) | 0.544 |
| *Dyslipidemia* | 3 (20%) | 1 (7%) | 3 (20%) | 0.508 |
| *Chronic cardiac failure* | 1 (7%) | 0 (0%) | 0 (0%) | 0.360 |
| *Cerebrovascular disease* | 2 (13%) | 0 (0%) | 0 (0%) | 0.123 |
| *Chronic Kidney Failure* | 1 (7%) | 0 (0%) | 0 (0%) | 0.360 |
| *Hypo- or hyperthyroidism* | 1 (7%) | 1 (7%) | 3 (20%) | 0.407 |

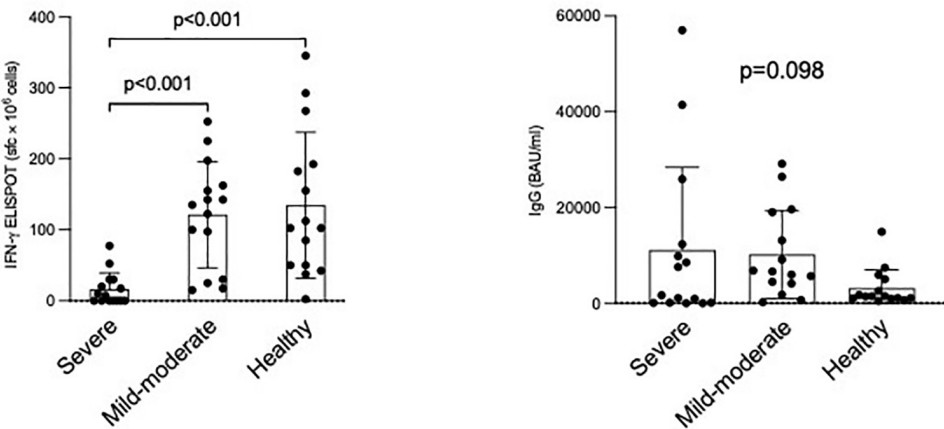

**Fig 1. The frequencies of spike-specific IFN-γ releasing cells and the titers of IgG measured in our cohorts of cases are depicted on the left and right, respectively.** The bottom and top of the box indicate the 25th and 75th percentiles, respectively. The horizontal band close to the middle of the box represents the median, whereas the ends of the whiskers represent the 10th and 90th percentiles. P values within study cohorts are also reported.

significantly lower in SARS-CoV-2 severe patients requiring hospitalization (8 [0; 26] s.f. c.×10⁶), as compared to COVID-19 mild-moderate patients (135 [64; 159] s.f.c.×10⁶; p<0.001) and healthy subjects (103 [50; 188] s.f.c.×10⁶; p<0.001). No difference was recorded between mild-moderate patients and healthy subjects (p = 0.852). Conversely, the anti-Spike protein IgG (p = 0.098) levels were similar among the three cohorts of cases. In particular, IgG (BAU/ml) titers were 1670 [124; 12300] in severe patients, 6600 [4100; 19000] in mild-moderate

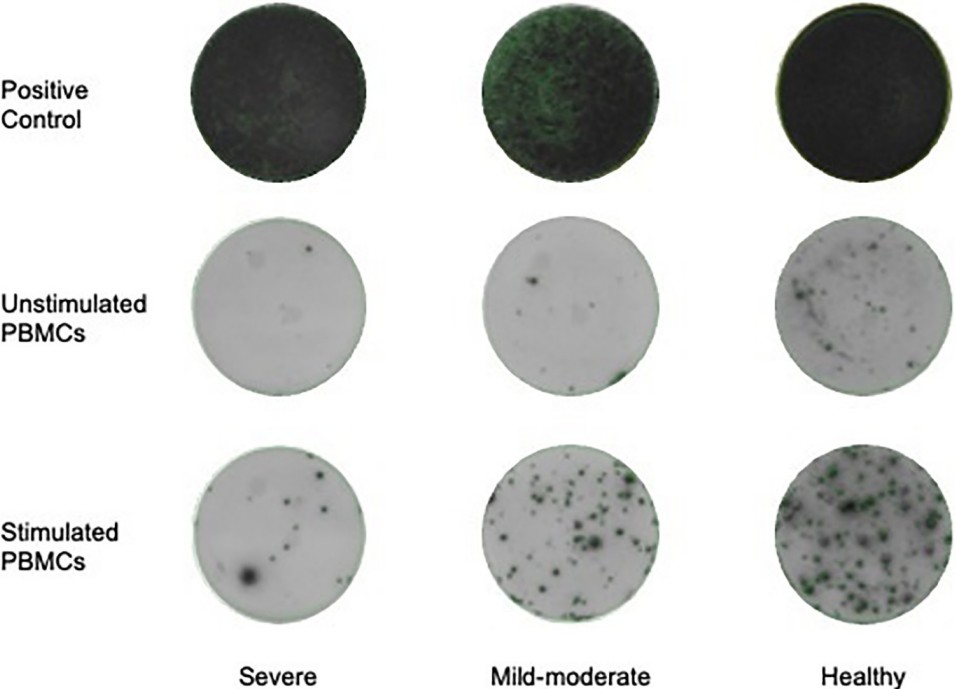

**Fig 2. The figure reports a representative IFN-γ ELISPOT from one case per severe, mild-moderate and healthy cohorts, from the left to the right.**

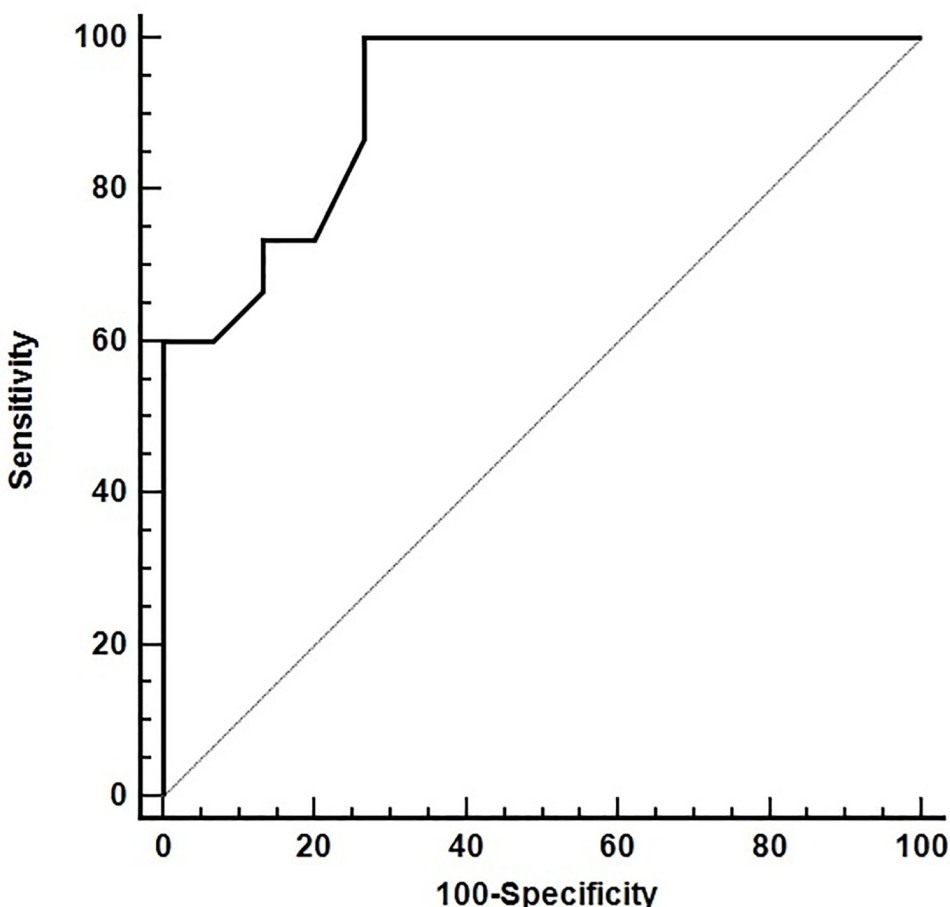

**Fig 3. The figure depicts the Receiver Operating Curve (ROC) of the rate of spike-specific IFNγ-releasing to discriminate mild-moderate from severe patients.**

patients and 1470 [1020; 5040] in healthy subjects. All cases but one showed positive IgG levels. All 45 cases had an IgM titer below the positivity threshold and under the analytic sensitivity of the test and therefore they cannot be reported for quantitative comparisons.

The ROC analysis indicated the rate of spike-specific IFNγ-releasing cells was a good parameter at discriminating correctly severe from mild-moderate patients (area under the curve: 0.9289; 95% CI: 0.8376–1.000; $p < 0.0001$). A diagnostic specificity of 100% was observed for s.f.c. $> 81.2/10^6$ PBMCs, a value corresponding to a diagnostic sensitivity of 66.7% (Fig 3).

## Discussion

In this exploratory study of subjects with two doses of the BNT162b2 vaccine, we report that the frequencies of spike-specific IFNγ-releasing cells (i.e. the pattern of T cell response) in severe COVID-19 patients is lower than in mild-moderate patients after breakthrough SARS-CoV-2 infection and in healthy subjects. Differently, the humoral response does not differ among our three cohorts. Considering our findings, frequency of spike-specific IFNγ-releasing cells appears to be a relevant marker for severe disease risk that larger prospective studies should confirm. Other studies investigated this area, most of which looked at humoral immunological parameters, such as specific SARS-CoV-2 antibodies [16] and cytokines [20], or

phenotypic characterization of lymphocyte subsets [20, 21]. The frequency of IFNγ-releasing cells is a less widespread approach, mainly due to the methodological complexity of the measurements.

Our findings are of paramount importance: in case of SARS-CoV-2 infection, the rate of IFNγ-releasing cells in dually vaccinated patients requiring hospitalization is significantly lower than patients with a pauci- or asymptomatic manifestation. ROC analysis demonstrated good diagnostic accuracy for marker T cell response, which means that it is possible to define cut-off values that can optimize diagnostic specificity or sensitivity.

Our study substantially confirms previous findings reported in unvaccinated patients. During the first wave of SARS-CoV-2 epidemic, Chandran et al. reported that the immune system reacts to SARS-CoV-2 infection with a rapid and efficient T cell response in non-severe COVID-19 patients [22]. Tan et al. reported that the early assessment of IFNγ-releasing cells was positively related with the severity and the course of the disease in 12 patients [23]. In particular, in severe COVID-19 patients the low (or even absent) quantity of IFNγ-releasing T cells in the peripheral blood could be caused by a defective induction of SARS-CoV-2 T cells [23]. Furthermore, T cell response to SARS-CoV-2 infection varies along the disease course, increasing progressively during the first 15 days after symptoms onset [23]. Of note, our study population tested positive to SARS-CoV-2 and/or showed flu-like symptoms within 36 hours from study inclusion, zeroing the possible bias of different T cell response related to the disease progression.

The role of IFNγ-releasing T cells in the immune response against SARS-CoV-2 have been extensively assessed [24]. A high IFNγ-producing T-cell activity is strongly associated with a low disease severity in acute [25] and in convalescent COVID-19 patients [6, 8, 26]. These results from unvaccinated patients are similar to our findings in vaccinated patients.

In vaccinated and severe patients, the lack of a T-cell mediated response remains to be clarified if it is associated to a reduced response of the patient to the vaccine (and therefore already present before SARS-CoV-2 infection) or if it is an inappropriate and low reaction directly to the infection. A recent study reported that dually vaccinated patients, admitted to Intensive Care Unit, lack of T-cell response despite high IgG titers [16]. The authors raised concerns regarding the assessment of the sole humoral immune response, suggesting the need to also assess the cellular response to SARS-CoV-2 at least in high-risk patients [16]. Therefore, we can formally support the speculation that the T-cell response "provides the underpinning control of serious tissue damage", despite the presence of high IgG titers [13]. In addition, our findings suggest that the lack of a T-cell mediated response may identify patients at risk of severe disease in case of SARS-CoV-2 infection, thus benefiting of a "booster" dose of vaccine. If an uncoordinated T-cell activity and antibody responses expose vaccinated subjects to the risk of severe disease, these patients would benefit of a "booster dose" before being SARS-CoV-2 infected [16]; otherwise, after infection, the immunological status may only predict the quick development of a severe disease requiring hospitalization. Of note, this hypothesis requires further specifically designed investigations to be confirmed or not.

We also titer the IgM and IgG levels to evaluate the humoral immune response. All patients showed a negative titer of IgM, well below the analytic sensitivity of the test, and a dominance of IgG, as expected by the IgG class-switch of spike-specific memory B cells [27]. Previous studies reported that after two doses of BNT162b2 mRNA vaccine the mean antibody titer reduces over time; in particular, although in all subjects the authors recorded a high level of antibodies after three months from the vaccination [28], in a following analysis the antibodies' titer declined over time with very low titers in subjects which received their second dose more than 150 days before [29]. In keeping with these findings [28, 29], we also recorded a detectable IgG titer in all included patients, although they received their last dose more than 150 days before. Of note, IgG titers do not correlate with the disease severity and patients' outcomes

[30], supporting our hypothesis that the cellular immune response plays a major role at this regard.

Before drawing our conclusions, some limitations deserve discussion. First, the patients' sample might be considered small. However, this sample is even larger than a previously published study on a similar topic [16]. In addition, this study has a merely explorative aim. Second, we did not assess the possible degree of peripheral lymphopenia and characterize in detail the different response of CD4+ and CD8+ T-cells. Of note, in more severe and critically COVID-19 patients, counts of peripheral CD4+ and CD8+ T cells are reduced, while their status is hyperactivated [31]. Third, our study included patients receiving a dual dose of vaccine against SARS-CoV-2; however, it remains unclear and to be investigated if a similar cell immune response verifies also in patients which received a "booster" dose. Fourth, we investigated the Spike-specific T cell responses but not to other SARS-CoV-2 structural (membrane and nucleoprotein) and non-structural (open reading frames) proteins. Since the T cell response might be multi-antigenic and it may play a role in understanding different disease severity [9, 25], further studies specifically designed are required. Last, included patients with severe COVID-19 disease were older than both mild-moderate patients and healthy subjects. Even if not statistically significant, this should be considered as a possible confounder.

## Conclusions

In conclusion, patients vaccinated with 2-dose of vaccine and developing a severe form of COVID-19 show a lower cellular-mediated immune response than mild-moderate or healthy vaccinated subjects, although antibody titers were similar.

## Author Contributions

**Conceptualization:** Eugenio Garofalo, Flavia Biamonte, Camillo Palmieri, Carlo Torti, Francesco Saverio Costanzo, Federico Longhini, Andrea Bruni.

**Data curation:** Eugenio Garofalo, Flavia Biamonte, Camillo Palmieri, Anna Martina Battaglia, Alessandro Sacco, Eugenio Biamonte, Giuseppe Neri, Giulio Cesare Antico, Serafina Mancuso, Giuseppe Foti, Carlo Torti, Francesco Saverio Costanzo, Federico Longhini, Andrea Bruni.

**Formal analysis:** Eugenio Garofalo, Flavia Biamonte, Camillo Palmieri, Anna Martina Battaglia, Alessandro Sacco, Eugenio Biamonte, Giuseppe Neri, Giulio Cesare Antico, Serafina Mancuso, Giuseppe Foti, Carlo Torti, Francesco Saverio Costanzo, Federico Longhini, Andrea Bruni.

**Investigation:** Eugenio Garofalo, Flavia Biamonte, Camillo Palmieri, Anna Martina Battaglia, Alessandro Sacco, Eugenio Biamonte, Giuseppe Neri, Giulio Cesare Antico, Serafina Mancuso, Giuseppe Foti, Carlo Torti, Francesco Saverio Costanzo, Federico Longhini, Andrea Bruni.

**Methodology:** Eugenio Garofalo, Flavia Biamonte, Camillo Palmieri, Anna Martina Battaglia, Alessandro Sacco, Eugenio Biamonte, Giuseppe Neri, Giulio Cesare Antico, Serafina Mancuso, Giuseppe Foti, Carlo Torti, Francesco Saverio Costanzo, Federico Longhini, Andrea Bruni.

**Supervision:** Federico Longhini.

**Writing – original draft:** Eugenio Garofalo, Flavia Biamonte, Camillo Palmieri, Carlo Torti, Francesco Saverio Costanzo, Federico Longhini, Andrea Bruni.

**Writing – review & editing:** Anna Martina Battaglia, Alessandro Sacco, Eugenio Biamonte, Giuseppe Neri, Giulio Cesare Antico, Serafina Mancuso, Giuseppe Foti, Carlo Torti, Francesco Saverio Costanzo, Federico Longhini.

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
