## [Decision Letter · Decision Letter 0]

25 Oct 2022

PONE-D-22-23169Hospitalized and non-hospitalized SARS-CoV-2 vaccinated patients show different frequencies of IFNγ-releasing cells: an exploratory studyPLOS ONE

Dear Dr. Longhini,

Thank you for submitting your manuscript to PLOS ONE. After careful consideration, we feel that it has merit but does not fully meet PLOS ONE’s publication criteria as it currently stands. Therefore, we invite you to submit a revised version of the manuscript that addresses the points raised during the review process.

We look forward to receiving your revised manuscript.

Kind regards,

Mohd Adnan, PhD

Academic Editor

PLOS ONE

Journal Requirements:

2. You indicated that you had ethical approval for your study. Please clarify whether minors (participants under the age of 18 years) were included in this study. If yes, in your Methods section, please ensure you have also stated whether you obtained consent from parents or guardians of the minors included in the study or whether the research ethics committee or IRB specifically waived the need for their consent.

3. I would appear that critically ill patients were included in your study. Please describe in your methods section how capacity to provide consent was determined for the participants in this study. Please also state whether your ethics committee or IRB approved this consent procedure. If you did not assess capacity to consent please briefly outline why this was not necessary in this case.

5.We note that you have indicated that data from this study are available upon request. PLOS only allows data to be available upon request if there are legal or ethical restrictions on sharing data publicly. For more information on unacceptable data access restrictions, please see http://journals.plos.org/plosone/s/data-availability#loc-unacceptable-data-access-restrictions.

Additional Editor Comments:

Reviewers have commented against the acceptance of manuscript in its current form. Manuscript suffers from serious concerns regarding the implemented protocol as well as presentation of the data. Please revise in light of reviewers comments and resubmit accordingly.

Reviewers' comments:

Reviewer's Responses to Questions

**Comments to the Author**

1. Is the manuscript technically sound, and do the data support the conclusions?

Reviewer #1: Yes

Reviewer #2: Yes

Reviewer #3: Partly

2. Has the statistical analysis been performed appropriately and rigorously? 

Reviewer #1: Yes

Reviewer #2: Yes

Reviewer #3: Yes

3. Have the authors made all data underlying the findings in their manuscript fully available?

Reviewer #1: Yes

Reviewer #2: Yes

Reviewer #3: Yes

4. Is the manuscript presented in an intelligible fashion and written in standard English?

Reviewer #1: Yes

Reviewer #2: Yes

Reviewer #3: Yes

5. Review Comments to the Author

Reviewer #1: I'd like to thank you for asking me to review this interesting paper which explains that 2-doses vaccinated patients requiring hospitalization for COVID-19 show a cellular-mediated immune response lower than non-hospitalized or controls, despite similar antibody titers.

The authors should read these published articles to deep discuss their findings:

-Coppeta L, Ferrari C, Somma G, Mazza A, D'Ancona U, Marcuccilli F, Grelli S, Aurilio MT, Pietroiusti A, Magrini A, Rizza S. Reduced Titers of Circulating Anti-SARS-CoV-2 Antibodies and Risk of COVID-19 Infection in Healthcare Workers during the Nine Months after Immunization with the BNT162b2 mRNA Vaccine. Vaccines (Basel). 2022 Jan 18;10(2):141. doi: 10.3390/vaccines10020141. PMID: 35214600; PMCID: PMC8879462.

-Coppeta L, Somma G, Ferrari C, Mazza A, Rizza S, Trabucco Aurilio M, Perrone S, Magrini A, Pietroiusti A. Persistence of Anti-S Titre among Healthcare Workers Vaccinated with BNT162b2 mRNA COVID-19. Vaccines (Basel). 2021 Aug 25;9(9):947. doi: 10.3390/vaccines9090947. PMID: 34579184; PMCID: PMC8472926.

-Sahin U., Muik A., Vogler I. BNT162b2 Induces SARS-CoV-2-Neutralising Antibodies and T Cells in Humans. Preprint. [(accessed on 11 December 2020)]. Available online: https://www.medrxiv.org/content/10.1101/2020.12.09.20245175v1

Reviewer #2: Thank you for the opportunity to review this interesting manuscript. The authors describe the immune response of hospitalized and ambulatory patients infected with COVID-19 and use healthy individuals as controls. Their main finding was a low level of IFNγ-releasing cells among hospitalized patients and a high predicting value for severe disease. While their findings are of value for future research and detecting patients at risk, there are still some major issues that should be addressed.

Major issues:

#1: Definition of study groups: the authors define the groups as hospitalized and non-hospitalized. However, there are several reasons for admission of COVID-19 patients beside their disease severity. I recommend changing the groups (throughout the paper, title, and conclusion) to severe, mild-moderate and healthy subjects.

#2: Study design: For proper validation of the results and the ability to implement them in future studies, several important issues must be addressed by the authors:

Methods:

- How were patients selected? Was it during their stay in the emergency department?

- At which point in the disease course the blood samples were drawn – was it at admission to the COVID-19 department?

- Did patients with severe disease received dexamethasone (or other steroids) for their disease prior to their inclusion in the study? If so – for how many days?

Results:

- What is the mean time from the vaccines to study inclusion for each group?

- What is the mean time from the PCR and from the first COVID-related symptoms to study inclusion for each group?

- Were any patients with mild/moderate disease deteriorated after being included in the study?

- Any significant comparison between two of the three groups should be addressed in Table 1. Currently the footnote of the table addresses this issue, but it is not mentioned in the table.

#3: discussion: While this section is properly written and presents the main findings, I believe the authors need to further discuss the following ideas:

- What is your theory for the basis of your results?

Do you think the low levels of IFNγ-releasing cells were present in these patients before being infected with COVID-19, which means it can be screened in the wider vaccinated population? On the other hand, do you think these patients had an inappropriate response to the infection (which reflects in low levels of IFNγ-releasing cells, and is also the basis for the study you cite in citation 12) and therefore have severe disease? If so, screening is relevant only among infected patients.

Based on your theory you should relate to previous studies, to the results of the healthy subjects (and their high levels of IFNγ-releasing cells) and to the possible implications in the general population.

- Previous studies describe both high and low levels of IFNγ among patients with severe disease, while this was not discussed by the authors in relation to their findings.

- Levels of IgG-S were previously shown not to correlate with disease severity and outcomes. The authors should address it to support their findings. In this regard, I recommend them to use the following paper which showed similar results for comparison: https://doi.org/10.1371/journal.pone.0268050

- The age of hospitalized patients was substantially older (p value is not everything when it’s a small number of patients). The authors should address this issue as a possible confounder or limitation.

Minor issues:

– In the end of the results, you state “A diagnostic specificity of 100% was observed for s.f.c. > 81.2 x 106…”. If I understand correctly, a lower value than 81.2 is indicative for a severe disease not higher. If so it should be changed accordingly.

Reviewer #3: Manuscript Title: Hospitalized and non-hospitalized SARS-CoV-2 vaccinated patients show different frequencies of IFNγ-releasing cells: an exploratory study

Summary:

This is a short research article that showed that hospitalized (severe) COVID-19 patients (n=15) present significantly lower level of Spike-specific T cells as compared to mild COVID-19 patients (n=15) and uninfected vaccinated controls (n=15) after breakthrough infection. In comparison, there is no significant differences for the IgG antibody titre among the three groups.

While the manuscript is simple, it might add some light on importance of cellular immunity in controlling SARS-CoV-2 infection. However there are large limitations that should be clearly highlighted.

First, this is certainly not the first manuscript that analyze T cell response in patients with severe or mild or asymptomatic SARS-CoV-2 infection. As such the introduction should acknowledge the work on T cell response published by other authors that have already showed presence of T cells not only in Covid-19 but also on asymptomatic individuals ( i.e Sekine, T. et al. Robust T Cell Immunity in Convalescent Individuals with Asymptomatic or Mild COVID-19. Cell 183, 158-168.e14 (2020), Le Bert, N. et al. Highly functional virus-specific cellular immune response in asymptomatic SARS-CoV-2 infection. J Exp Med 218, e20202617 (2021).). Furthermore, the abstract ( and discussion) does does not clarify that the analysis of Spike-specific T cell response was done after infection and not before the infection. As such the work does not provide any data about” correlate of protection from infection” but only analyze the pattern of T cell response after SARS-CoV-2 infection. This should be clearly mentioned.

As such the abstract should be changed . Authors should clarify that the analysis was done after infection. Thus “ In this study we aim to evaluate the spike-specific immune responses in patients requiring or not hospitalization for SARS-CoV-2 infection, after breakthrough infection following two doses of BNT162b2 mRNA vaccine” .

In addition the authors should also acknowledge that their work substantially confirmed in vaccinated individuals previous work that show lower level of SARS-CoV-2 specific T cells in individuals with severe COVID-19 ( I.e. Tan, A., et al. 2021. Early induction of functional SARS-CoV-2-specific T cells associates with rapid viral clearance and mild disease in COVID-19 patients. Cell Reports, 34(6), p.108728. and Chandran, A. et al. Rapid synchronous type 1 IFN and virus-specific T cell responses characterize first wave non-severe SARS-CoV-2 infections. Cell Reports Medicine 3, 100557–100557 (2022).) ).

The authors should also discuss and pointed out that in severe COVID-19, lymphocytes count is decreasing and SARS-CoV-2 T cells can be recruited at the site of inflammation and as such the quantity of circulating SARS-CoV-2 T cells can be reduced. This is why it is important to perform longitudinal early analysis ( as in Tan, A., et al. 2021. Early induction of functional SARS-CoV-2-specific T cells associates with rapid viral clearance and mild disease in COVID-19 patients. Cell Reports, 34(6), p.108728. and Chandran, A. et al. Rapid synchronous type 1 IFN and virus-specific T cell responses characterize first wave non-severe SARS-CoV-2 infections. Cell Reports Medicine 3, 100557–100557 (2022) and not only at single time when inflammatory events might be at their peak. This is why is also important to understand when their T cell analysis was performed .

The authors should also acknowledge that the analysis was focused only on Spike, but since samples were collected from donors who were infected with SARS-CoV-2, instead of looking only at Spike-specific T cell responses, it will be complimentary to also look T cells specific for other SARS-CoV-2 structural (Membrane and Nucleoprotein) and non-structural (ORFs) proteins. This will allow the authors to not only better understand the breath of the T cell responses, which is possibly different among the different cohorts, but also know the better representative magnitude of SARS-CoV-2 T cell response, as cellular immunity is contributed by more than Spike-specific T cells. Multi-antigenic T cell response might play a paramount role in preventing severe COVID-19, so it should be discussed in the context of this study.

Other minor points:

1. The authors should be clearer about the demographics of the patients recruited, particularly the number of days post-infection, number of days post-vaccination and whether the donors have recovered from COVID-19, as they were not clear on whether they are studying acute or convalescent samples.

2. Could the authors clarify on the use of steroid therapy (the statement “since more than 10 days”) or other therapy (i.e. antivirals or antibody), particularly in the “severe/critical COVID-19” cohort.

3. The study could benefit from the availability of clinical parameters describing the patients (degree of peripheral lymphopenia in hospitalized vs non-hospitalized patients, inflammatory markers, etc.)

4. It might be better for the authors to include individual datapoints on top of the box plots for the graphs plotted in Figure 1. As it seems like there are obvious outliers, and that cannot be easily inferred with the current graphs. Plotting individual datapoints will allow the readers to better appreciate the data.

6. PLOS authors have the option to publish the peer review history of their article (what does this mean?). If published, this will include your full peer review and any attached files.

Reviewer #1: No

Reviewer #2: No

Reviewer #3: No

---

## [Author Response · Author response to Decision Letter 0]

8 Dec 2022

Reviewer #1

The authors should read these published articles to deep discuss their findings:

- Coppeta L, Ferrari C, Somma G, Mazza A, D'Ancona U, Marcuccilli F, Grelli S, Aurilio MT, Pietroiusti A, Magrini A, Rizza S. Reduced Titers of Circulating Anti-SARS-CoV-2 Antibodies and Risk of COVID-19 Infection in Healthcare Workers during the Nine Months after Immunization with the BNT162b2 mRNA Vaccine. Vaccines (Basel). 2022 Jan 18;10(2):141. doi: 10.3390/vaccines10020141. PMID: 35214600; PMCID: PMC8879462.

- Coppeta L, Somma G, Ferrari C, Mazza A, Rizza S, Trabucco Aurilio M, Perrone S, Magrini A, Pietroiusti A. Persistence of Anti-S Titre among Healthcare Workers Vaccinated with BNT162b2 mRNA COVID-19. Vaccines (Basel). 2021 Aug 25;9(9):947. doi: 10.3390/vaccines9090947. PMID: 34579184; PMCID: PMC8472926.

- Sahin U., Muik A., Vogler I. BNT162b2 Induces SARS-CoV-2-Neutralising Antibodies and T Cells in Humans. Preprint. [(accessed on 11 December 2020)]. Available online: https://www.medrxiv.org/content/10.1101/2020.12.09.20245175v1

We thank the Reviewer for his/her suggestion. We included and quoted the two papers by Coppeta et al. in the Discussion section, whereas we prefer to not include the paper by Sahin, since this is a non-peer reviewed paper published as a pre-print. Therefore, this manuscript may contain some flaws requiring the revision process before publication and to consider it as scientifically valid manuscript.

 

Reviewer #2

Major issues:

1) Definition of study groups: the authors define the groups as hospitalized and non-hospitalized. However, there are several reasons for admission of COVID-19 patients beside their disease severity. I recommend changing the groups (throughout the paper, title, and conclusion) to severe, mild-moderate and healthy subjects.

We thank the Reviewer for his/her suggestion. We have completely followed her/his suggestion and cohorts have been renamed as required. 

2) Study design: For proper validation of the results and the ability to implement them in future studies, several important issues must be addressed by the authors:

Methods:

- How were patients selected? Was it during their stay in the emergency department?

We have now included in the revised version of the manuscript this information.

- At which point in the disease course the blood samples were drawn – was it at admission to the COVID-19 department?

As already mentioned in the manuscript: “After inclusion, demographic and clinical features were recorded. Furthermore, blood samples were collected in BD Vacutainer plasma tubes containing lithium heparin tubes (Becton Dickinson; Plymouth, UK).”

- Did patients with severe disease received dexamethasone (or other steroids) for their disease prior to their inclusion in the study? If so – for how many days?

No patients received steroids before study inclusion. This is now specified in the results section.

Results:

- What is the mean time from the vaccines to study inclusion for each group?

The reviewer is right, we missed this information that is relevant. We have now included it in Table 1. Since we assumed data as non-parametric, we have described the time (i.e. days) from last vaccine dose in median [interquartile range].

- What is the mean time from the PCR and from the first COVID-related symptoms to study inclusion for each group?

Thanks, we have now specified this in Materials and Methods. Patients were classified in the 3 cohorts at the time of swab result that corresponds to the inclusion time.

- Were any patients with mild/moderate disease deteriorated after being included in the study?

None of the included mild/moderate patients deteriorated to severe after study inclusion. This is now specified in the revised text.

- Any significant comparison between two of the three groups should be addressed in Table 1. Currently the footnote of the table addresses this issue, but it is not mentioned in the table.

We thank the Reviewer for her/his indication. There are no significant differences among populations and footnotes have been deleted accordingly. 

#3: discussion: While this section is properly written and presents the main findings, I believe the authors need to further discuss the following ideas:

- What is your theory for the basis of your results?

Do you think the low levels of IFNγ-releasing cells were present in these patients before being infected with COVID-19, which means it can be screened in the wider vaccinated population? On the other hand, do you think these patients had an inappropriate response to the infection (which reflects in low levels of IFNγ-releasing cells, and is also the basis for the study you cite in citation 12) and therefore have severe disease? If so, screening is relevant only among infected patients.

Based on your theory you should relate to previous studies, to the results of the healthy subjects (and their high levels of IFNγ-releasing cells) and to the possible implications in the general population.

We thank the Reviewer for his/her comment. We have included in the manuscript (discussion) this concept that is trivial. However, it remains difficult to state if our findings are related to the infection or a weak response to the vaccine. At this regard, we have left open the possibility and the question to a specifically designed trial.

- Previous studies describe both high and low levels of IFNγ among patients with severe disease, while this was not discussed by the authors in relation to their findings.

Following the Reviewer’s suggestion, we have included a paragraph in the discussion.

- Levels of IgG-S were previously shown not to correlate with disease severity and outcomes. The authors should address it to support their findings. In this regard, I recommend them to use the following paper which showed similar results for comparison: https://doi.org/10.1371/journal.pone.0268050

We thank the Reviewer for her/his thoughtful suggestion that has been fully included in the discussion.

- The age of hospitalized patients was substantially older (p value is not everything when it’s a small number of patients). The authors should address this issue as a possible confounder or limitation.

Done, thanks.

Minor issues:

– In the end of the results, you state “A diagnostic specificity of 100% was observed for s.f.c. > 81.2 x 106…”. If I understand correctly, a lower value than 81.2 is indicative for a severe disease not higher. If so it should be changed accordingly.

In our study, diagnostic specificity refers to the IFN-γ ELISPOT's ability to correctly define patients with mild / moderate disease, which is 100% for s.f.c. greater than 81.2/PBMCs 106. 

Reviewer #3

First, this is certainly not the first manuscript that analyze T cell response in patients with severe or mild or asymptomatic SARS-CoV-2 infection. 

As such the introduction should acknowledge the work on T cell response published by other authors that have already showed presence of T cells not only in Covid-19 but also on asymptomatic individuals ( i.e Sekine, T. et al. Robust T Cell Immunity in Convalescent Individuals with Asymptomatic or Mild COVID-19. Cell 183, 158-168.e14 (2020), Le Bert, N. et al. Highly functional virus-specific cellular immune response in asymptomatic SARS-CoV-2 infection. J Exp Med 218, e20202617 (2021).). Furthermore, the abstract (and discussion) does not clarify that the analysis of Spike-specific T cell response was done after infection and not before the infection. As such the work does not provide any data about” correlate of protection from infection” but only analyze the pattern of T cell response after SARS-CoV-2 infection. This should be clearly mentioned.

Following the Reviewer suggestions, we have accordingly revised the manuscript and all suggestions have been included. 

As such the abstract should be changed. Authors should clarify that the analysis was done after infection. Thus “In this study we aim to evaluate the spike-specific immune responses in patients requiring or not hospitalization for SARS-CoV-2 infection, after breakthrough infection following two doses of BNT162b2 mRNA vaccine”.

Done.

2. In addition the authors should also acknowledge that their work substantially confirmed in vaccinated individuals previous work that show lower level of SARS-CoV-2 specific T cells in individuals with severe COVID-19 ( I.e. Tan, A., et al. 2021. Early induction of functional SARS-CoV-2-specific T cells associates with rapid viral clearance and mild disease in COVID-19 patients. Cell Reports, 34(6), p.108728. and Chandran, A. et al. Rapid synchronous type 1 IFN and virus-specific T cell responses characterize first wave non-severe SARS-CoV-2 infections. Cell Reports Medicine 3, 100557–100557 (2022).) 

Following the Reviewer suggestions, we have accordingly revised the manuscript.

3. The authors should also discuss and pointed out that in severe COVID-19, lymphocytes count is decreasing, and SARS-CoV-2 T cells can be recruited at the site of inflammation and as such the quantity of circulating SARS-CoV-2 T cells can be reduced. This is why it is important to perform longitudinal early analysis ( as in Tan, A., et al. 2021. Early induction of functional SARS-CoV-2-specific T cells associates with rapid viral clearance and mild disease in COVID-19 patients. Cell Reports, 34(6), p.108728. and Chandran, A. et al. Rapid synchronous type 1 IFN and virus-specific T cell responses characterize first wave non-severe SARS-CoV-2 infections. Cell Reports Medicine 3, 100557–100557 (2022) and not only at single time when inflammatory events might be at their peak. This is why is also important to understand when their T cell analysis was performed .

Following the Reviewer suggestions, we have accordingly revised the manuscript.

The authors should also acknowledge that the analysis was focused only on Spike, but since samples were collected from donors who were infected with SARS-CoV-2, instead of looking only at Spike-specific T cell responses, it will be complimentary to also look T cells specific for other SARS-CoV-2 structural (Membrane and Nucleoprotein) and non-structural (ORFs) proteins. This will allow the authors to not only better understand the breath of the T cell responses, which is possibly different among the different cohorts, but also know the better representative magnitude of SARS-CoV-2 T cell response, as cellular immunity is contributed by more than Spike-specific T cells. Multi-antigenic T cell response might play a paramount role in preventing severe COVID-19, so it should be discussed in the context of this study.

We agree with the Reviewer, and we acknowledge her/his concern as limitation of the study. Unfortunately, we do not have samples anymore to provide this adjunctive analysis. In addition, this was out of our study aim and therefore we now suggest that further studies are required at this regard.

Other minor points:

1. The authors should be clearer about the demographics of the patients recruited, particularly the number of days post-infection, number of days post-vaccination and whether the donors have recovered from COVID-19, as they were not clear on whether they are studying acute or convalescent samples.

As also required by Reviewer #2, we have included these data in the revised text.

2. Could the authors clarify on the use of steroid therapy (the statement “since more than 10 days”) or other therapy (i.e. antivirals or antibody), particularly in the “severe/critical COVID-19” cohort.

As also required by Reviewer #2, we have now specified that none of included patients received steroids before study inclusion 

3. The study could benefit from the availability of clinical parameters describing the patients (degree of peripheral lymphopenia in hospitalized vs non-hospitalized patients, inflammatory markers, etc.)

We agree with the Reviewer, but these data are not available for non-hospitalized (mild-moderate) patients and control (healthy) group. This is now recognized as limitation of the study.

4. It might be better for the authors to include individual datapoints on top of the box plots for the graphs plotted in Figure 1. As it seems like there are obvious outliers, and that cannot be easily inferred with the current graphs. Plotting individual datapoints will allow the readers to better appreciate the data.

Following the Reviewer’s suggestion, we have not plotted also individual datapoints.

---

## [Decision Letter · Decision Letter 1]

13 Jan 2023

PONE-D-22-23169R1Severe and mild-moderate SARS-CoV-2 vaccinated patients show different frequencies of IFNγ-releasing cells: an exploratory studyPLOS ONE

Dear Dr. Longhini,

Thank you for submitting your manuscript to PLOS ONE. After careful consideration, we feel that it has merit but does not fully meet PLOS ONE’s publication criteria as it currently stands. Therefore, we invite you to submit a revised version of the manuscript that addresses the points raised during the review process.

We look forward to receiving your revised manuscript.

Kind regards,

Mohd Adnan, PhD

Academic Editor

PLOS ONE

Journal Requirements:

Additional Editor Comments:

Manuscript is significantly improved by the authors. However, there are still some minor concerns raised by the reviewer. Please address these concerns and resubmit accordingly.

Reviewers' comments:

Reviewer's Responses to Questions

**Comments to the Author**

1. If the authors have adequately addressed your comments raised in a previous round of review and you feel that this manuscript is now acceptable for publication, you may indicate that here to bypass the “Comments to the Author” section, enter your conflict of interest statement in the “Confidential to Editor” section, and submit your "Accept" recommendation.

Reviewer #2: All comments have been addressed

Reviewer #3: All comments have been addressed

2. Is the manuscript technically sound, and do the data support the conclusions?

Reviewer #2: Yes

Reviewer #3: (No Response)

3. Has the statistical analysis been performed appropriately and rigorously? 

Reviewer #2: Yes

Reviewer #3: (No Response)

4. Have the authors made all data underlying the findings in their manuscript fully available?

Reviewer #2: Yes

Reviewer #3: (No Response)

5. Is the manuscript presented in an intelligible fashion and written in standard English?

Reviewer #2: No

Reviewer #3: (No Response)

6. Review Comments to the Author

Reviewer #2: Thank you for the opportunity to re-review this manuscript. The authors have made substantial changes and answered all the issues I've raised. The manuscript has substantially improved. Still there are several issues which should be addressed:

- Many minor spelling and grammar mistake should be revised. For example – in the abstract – few spaces are missing, "discriminating" should be discriminate, and so on. This occurs throughout the text.

- If I understand correctly after the revision – all included patients were diagnosed only upon hospital arrival and not before? If so you should change the sentences describing the first two groups which state "in the previous 10 days". Later you wrote that PCR was positive within 36 hours (results), please state the correct timing in all the places.

Reviewer #3: The authors have addressed my main reservations. The text is more balanced and describes objectively the results obtained.

7. PLOS authors have the option to publish the peer review history of their article (what does this mean?). If published, this will include your full peer review and any attached files.

Reviewer #2: No

Reviewer #3: **Yes: **Antonio Bertoletti

---

## [Author Response · Author response to Decision Letter 1]

16 Jan 2023

Journal Requirements:

References have been checked and they should be fine per style. To the best of our knowledge, no retracted manuscripts have been included.

Additional Editor Comments:

Manuscript is significantly improved by the authors. However, there are still some minor concerns raised by the reviewer. Please address these concerns and resubmit accordingly.

Concerns pointed out from Reviewer #2 have been addressed. Please, see below.

Reviewers' comments:

Reviewer #2:

Thank you for the opportunity to re-review this manuscript. The authors have made substantial changes and answered all the issues I've raised. The manuscript has substantially improved. Still there are several issues which should be addressed:

- Many minor spelling and grammar mistake should be revised. For example – in the abstract – few spaces are missing, "discriminating" should be discriminate, and so on. This occurs throughout the text.

Following the Reviewer’s comments, we have checked the entire manuscript and correct.

- If I understand correctly after the revision – all included patients were diagnosed only upon hospital arrival and not before? If so you should change the sentences describing the first two groups which state "in the previous 10 days". Later you wrote that PCR was positive within 36 hours (results), please state the correct timing in all the places.

We agree with the Reviewer that the manuscript lacked clarity. The need for a positive PCR swab in the previous 10 days was an inclusion criterion; however, in the results we have specified that all patients resulted positive within 36 hours before inclusion. We have decided to leave only the inclusion timing (36 hours) and to delete the time limit of inclusion criterion. We believe that now the manuscript has improved in its clarity.

Reviewer #3:

 The authors have addressed my main reservations. The text is more balanced and describes objectively the results obtained.

Ok, thanks.

---

## [Editor Report · Decision Letter 2]

24 Jan 2023

Severe and mild-moderate SARS-CoV-2 vaccinated patients show different frequencies of IFNγ-releasing cells: an exploratory study

PONE-D-22-23169R2

Dear Dr. Longhini,

We’re pleased to inform you that your manuscript has been judged scientifically suitable for publication and will be formally accepted for publication once it meets all outstanding technical requirements.

Kind regards,

Mohd Adnan, PhD

Academic Editor

PLOS ONE
---

## [Editor Report · Acceptance letter]

31 Jan 2023

PONE-D-22-23169R2 

Severe and mild-moderate SARS-CoV-2 vaccinated patients show different frequencies of IFNγ-releasing cells: an exploratory study 

Dear Dr. Longhini:

I'm pleased to inform you that your manuscript has been deemed suitable for publication in PLOS ONE. Congratulations! Your manuscript is now with our production department. 

Kind regards, 

on behalf of

Dr. Mohd Adnan 

Academic Editor

PLOS ONE